# Allosteric regulation alters carrier domain translocation in pyruvate carboxylase

Yumeng Liu [1], Melissa M. Budelier [1], Katelyn Stine[1] & Martin St. Maurice[1]

Pyruvate carboxylase (PC) catalyzes the ATP-dependent carboxylation of pyruvate to oxaloacetate. The reaction occurs in two separate catalytic domains, coupled by the long-range translocation of a biotinylated carrier domain (BCCP). Here, we use a series of hybrid PC enzymes to examine multiple BCCP translocation pathways in PC. These studies reveal that the BCCP domain of PC adopts a wide range of translocation pathways during catalysis. Furthermore, the allosteric activator, acetyl CoA, promotes one specific intermolecular carrier domain translocation pathway. These results provide a basis for the ordered thermodynamic state and the enhanced carboxyl group transfer efficiency in the presence of acetyl CoA, and reveal that the allosteric effector regulates enzyme activity by altering carrier domain movement. Given the similarities with enzymes involved in the modular synthesis of natural products, the allosteric regulation of carrier domain movements in PC is likely to be broadly applicable to multiple important enzyme systems.

[1] Department of Biological Sciences, Marquette University, Milwaukee, WI 53201, USA. Correspondence and requests for materials should be addressed to M.S.M. (email: martin.stmaurice@marquette.edu)

Pyruvate carboxylase (PC; EC 6.4.1.1) is a biotin-dependent enzyme present in a wide range of species, ranging from bacteria to mammals. In most species, PC is a multi-functional, homotetrameric enzyme with each subunit comprised of four domains: the biotin carboxylase (BC) domain, the carboxyltransferase (CT) domain, the biotin carboxyl carrier protein (BCCP) domain, and the allosteric domain (also known as the pyruvate tetramerization (PT) domain) (Fig. 1). The tetramer is arranged as a dimer of dimers, with two antiparallel subunits forming one layer of the tetramer and two perpendicular subunits forming the opposing layer (Fig. 1). An essential biotin cofactor, tethered by an amide linkage to a specific lysine side-chain on the BCCP domain, serves to carry a carboxyl group between substrates in the BC and CT domains. The overall reaction is catalyzed in two steps: first, at the BC active site, the carboxyl group from bicarbonate is transferred to the N1′ position of the biotin cofactor with concomitant cleavage of ATP to ADP and P$_i$; second, at the CT active site, the carboxyl group is transferred from carboxybiotin to pyruvate, to produce oxaloacetate. The two half-reactions are coupled by a ≥60 Å translocation of the BCCP domain between the BC and CT domain active sites, facilitating the net carboxyl group transfer from bicarbonate to pyruvate.

It has been determined that the BCCP domain can translocate within one layer of the tetramer, between the BC domain of its own subunit and the CT domain of the opposing, antiparallel subunit[1–3] (Fig. 2, movements a). The BCCP domain is connected to, but offset from, the rest of the subunit by a flexible linker comprised entirely of random coil. This permits the BCCP domain to access a wide range of positions within the tetramer, consistent with the low occupancy of this domain in several X-ray structures of PC[2,4,5]. Despite the broad positional flexibility inherent to the BCCP domain, no studies have systematically assessed whether it is capable of additional catalytically productive translocations (Fig. 2, movements b–d).

The PC-catalyzed carboxylation of pyruvate to oxaloacetate serves as an important anaplerotic reaction to replenish citric acid cycle intermediates lost to a variety of competing biosynthetic pathways, including carbohydrate, amino acid, and lipid biosynthesis. Consequently, PC is subject to various degrees of allosteric regulation, with acetyl CoA serving as an allosteric activator of most PC enzymes and L-aspartate acting as an allosteric inhibitor of microbial PC enzymes[6–9]. Steady-state and pre-steady-state kinetics in PC suggest that acetyl CoA activates the overall reaction by accelerating the rate-limiting HCO$_3^-$-dependent MgATP cleavage in the BC domain[10,11]. Interestingly, in the absence of acetyl CoA, the PC-catalyzed rate of ATP cleavage is significantly reduced while, at the same time, the efficiency of coupling between the reactions in the BC and CT domains is decreased due to high levels of abortive ATP cleavage[12]. The effect of acetyl CoA on the reaction coupling efficiency suggests that it alters more than just the rate of ATP cleavage in the BC domain. Acetyl CoA may also regulate intermediate transfer between active sites, possibly by altering the physical translocation of the BCCP domain[13].

As with acetyl CoA, allosteric inhibition by L-aspartate alters the rate of ATP cleavage in the BC domain but, unlike acetyl CoA, it does not alter the coupling efficiency between the individual half-reactions[8]. Thus, while L-aspartate binds competitively with respect to acetyl CoA[8,9], its inhibitory activity does not appear to influence intermediate transfer between active sites.

X-ray crystallographic studies of PC cloned from multiple source organisms have provided structural snapshots of the enzyme both in the presence and absence of acetyl CoA. PC structures reveal either a symmetric (*Staphylococcus aureus* PC[5,14]) or an asymmetric tetramer (*Rhizobium etli* PC[2,4]),

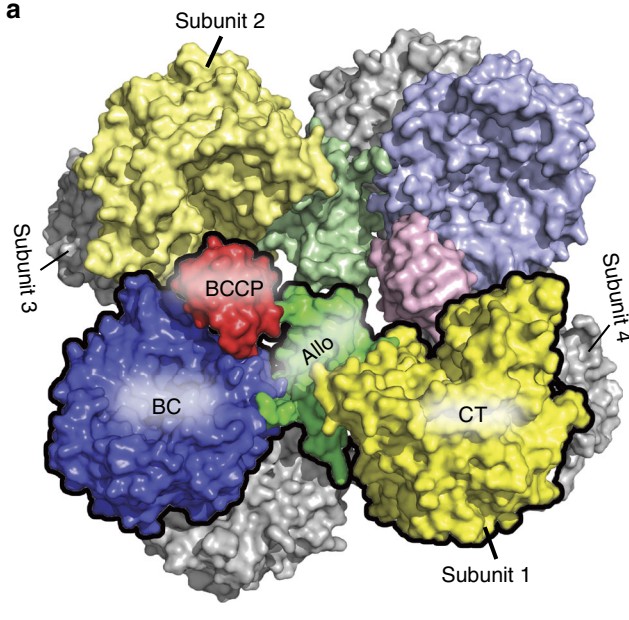

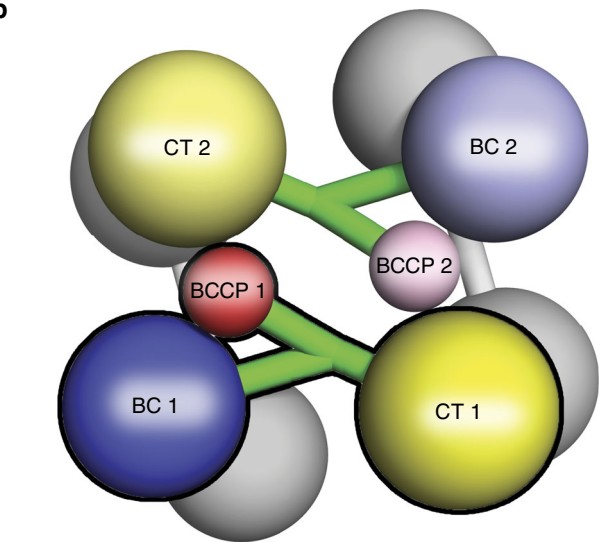

**Fig. 1** The quaternary structure of RePC. The subunits in RePC (2QF7) are organized to include two subunits on the upper layer and two subunits on the lower layer of the tetramer. The individual domains on the top layer are colored in blue for the biotin carboxylase (BC) domain; yellow for the carboxyltransferase (CT) domain; red for the biotin carboxyl carrier protein (BCCP) domain and green for the allosteric domain. The two subunits on the lower layer of the tetramer are colored entirely in gray for clarity. A space filling representation is shown in **a**, while a simplified cartoon representation of the subunit domains is shown in **b**

irrespective of whether acetyl CoA is present. Cryo-EM studies in SaPC more recently revealed that PC oscillates between a symmetrical and asymmetrical state during the catalytic cycle[15]. In SaPC, acetyl CoA was found to stabilize and promote BC domain dimerization through a subtle reorganization of the BC and allosteric (PT) domain dimers[5]. However, despite multiple structures, there is no evidence to support a significant remodeling of the domain organization or active site architecture in PC in response to acetyl CoA binding. Consequently, acetyl CoA is more likely to stimulate catalysis by altering dynamic processes, such as BCCP domain translocation.

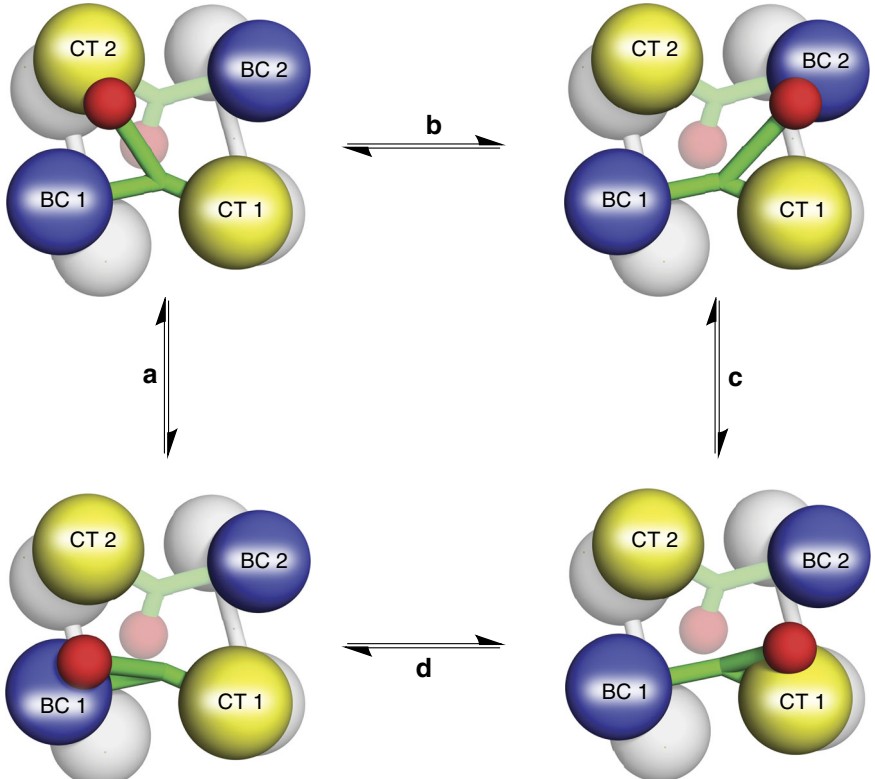

**Fig. 2** BCCP translocation pathways in PC. The BCCP domain is theoretically capable of four catalytically productive translocation pathways between the individual active sites of PC. Subunits 1 and 2 from the upper layer are colored according to their individual domains, while the lower layer is colored in grey for clarity. The four translocation pathways (**a–d**) are shown for the BCCP domain (colored in red) from subunit 1. For clarity, the BCCP domain from subunit 2 is displayed as partially transparent, but it is also capable of adopting these same four translocation pathways

Here, we use a series of mutated, hybrid PC enzymes to perform a detailed analysis of multiple translocation pathways for the BCCP domain during catalytic turnover in PC. We demonstrate that, surprisingly, the BCCP domain is capable of a much broader range of catalytically productive movements than previously recognized. We also describe the effect of allosteric effectors on BCCP domain translocation during catalysis by both *Aspergillus nidulans* PC (AnPC) and *R. etli* PC (RePC), two enzymes that respond very differently to acetyl CoA activation. We report that carrier domain translocation is regulated by acetyl CoA in RePC, an acetyl CoA sensitive PC enzyme. However, BCCP domain translocation is unaffected by the presence of the allosteric inhibitor, L-aspartate. These findings alter the classical description of catalysis in PC and suggest that carrier domain motions in swinging domain enzymes may be far more complex than previously imagined.

## Results

**Catalytic activity of ΔBC RePC dimers.** We previously demonstrated that a truncated construct of RePC, lacking the entire N-terminal BC domain (ΔBC RePC), is dimeric in solution and that it catalyzes the biotin-dependent, oxamate-induced decarboxylation of oxaloacetate nearly three times faster than wild-type RePC[4,16]. The translocation of the BCCP domain has been assumed to occur exclusively in trans, traveling from the BC domain on its own subunit to the CT domain on a neighboring subunit (i.e. Fig. 2, movement a)[2,15]. Given this assumption, the observed high rate of biotin-dependent oxaloacetate decarboxylation in ΔBC RePC was quite unexpected since the ΔBC RePC

dimer is incapable of forming the requisite intermolecular BCCP-CT domain interaction. Since a BCCP-CT domain interaction is absolutely required in order for this reaction to proceed[16], this result suggested that the BCCP domain is able to access the CT domain in an unexpected way, acting in *cis* to access the CT domain on its own subunit through an intramolecular interaction (i.e. Fig. 2, movement d).

To investigate whether the BCCP domain of the ΔBC RePC dimer can productively interact with the CT domain, the catalytic activities of two sets of ΔBC RePC heterodimers were analyzed. Two inactive constructs of ΔBC RePC were generated: one construct was mutated at the catalytically essential Thr882 of the CT domain (ΔBC T882A RePC), while a second construct was C-terminally truncated to eliminate the BCCP domain entirely (ΔBCΔBCCP RePC). Both of these mutations have previously been shown to result in a complete loss of activity in the oxamate-induced oxaloacetate decarboxylation reaction[16,17]. We attempted to generate pure heterodimers of ΔBC T882A and ΔBCΔBCCP RePC by co-expressing these with separate N-terminal affinity tags (polyhistidine and calmodulin (CaM)-binding peptide) and sequentially purifying them using Ni[2+]- and CaM-affinity resins. Unfortunately, the purified heterodimers quickly redistributed back into a mixed population of homo- and heterodimers (data not shown). To circumvent this problem, a population of hybrid dimers was generated by mixing together ΔBC T882A RePC with ΔBCΔBCCP RePC at a 1:1 ratio. Affinity co-purification was used to demonstrate that these dimers recombined into a fully heterogeneous population (Supplementary Fig. 1). Individually, neither of the mutated homodimers retained any catalytic

**Table 1 Oxaloacetate decarboxylation catalyzed by ΔBC dimers of RePC**

| RePC mutants | | $k_{cat}^{OIOD}$ (min$^{-1}$)[a] | $k_{cat}^{OD}$ (min$^{-1}$)[b] | biotin-dependent $k_{cat}$ (min$^{-1}$)[c] |
|---|---|---|---|---|
| [d]ΔBC RePC | | $10.2 \pm 0.3$[e] | $0.53 \pm 0.01$[e] | $9.7 \pm 0.3$ |
| ΔBC T882A | | $0.12 \pm 0.03$ | $0.07 \pm 0.03$ | $0.05 \pm 0.04$ |
| ΔBCΔBCCP | | $0.46 \pm 0.01$ | $0.45 \pm 0.01$[e] | $0.01 \pm 0.01$ |
| ΔBC T882A/ΔBCΔBCCP | | $1.30 \pm 0.05$ | $0.13 \pm 0.02$ | $1.2 \pm 0.1$ |

[a] $k_{cat}$ determined for oxaloacetate decarboxylation in the presence of 0.5 mM oxamate. The reported values are the average of three independent measurements from one purified sample. Errors are reported as the standard deviation
[b] $k_{cat}$ determined for oxaloacetate decarboxylation in the absence of oxamate. The reported values are the average of three independent measurements from one purified sample. Errors are reported as the standard deviation
[c] The biotin-dependent $k_{cat}$ is defined as $k_{cat}^{OIOD} - k_{cat}^{OD}$. Reported errors are propagated from the standard deviations
[d] Black shading indicates an inactive domain; light shading indicates an active domain
[e] From [16]

activity. However, biotin-dependent oxaloacetate decarboxylation activity was recovered when the two inactive homodimers were mixed together to form a population that included hybrid dimers (Table 1). This recovery of activity can only result from the BCCP domain accessing the CT domain on the opposing face of the ΔBC dimer, through an interaction that has never been recognized to be possible in PC.

To determine whether the BCCP domain on the ΔBC RePC dimer can act in *cis*, accessing the CT domain through an intramolecular interaction, we created a series of mixed heterodimer populations by combining a fully inactive ΔBCΔBCCP T882A RePC with a fully active ΔBC RePC. As above, the redistribution of purified heterodimers into a mixed population precluded an analysis of purified heterodimers. Instead, inactive ΔBCΔBCCP T882A RePC was mixed at increasing ratios with active ΔBC RePC as outlined in Fig. 3a, to alter the proportions of homodimers and heterodimers in the population. This approach generates wild-type ΔBC RePC homodimer that is increasingly diluted into heterodimers with fully inactive ΔBCΔBCCP T882A RePC. The heterodimers in the population will either be active, if the BCCP domain can act in *cis*, or inactive if the BCCP domain is unable to act in *cis*. The predicted curves for a system that acts 100 and 0% in *cis* are plotted in Fig. 3b, along with the relative activities measured for several mixed populations of ΔBC RePC and ΔBCΔBCCP T882A RePC in the presence and absence of acetyl CoA. From this curve, it is clear that wild-type ΔBC RePC acts primarily in *cis*, with the BCCP domain accessing the CT domain through an

intramolecular interaction. Together, these experiments demonstrate that, in the context of the ΔBC RePC dimer, the BCCP domain exhibits a much wider range of catalytically productive conformations than previously envisioned.

**Hybrid tetramers to probe BCCP translocation pathways.** The kinetic studies of heterodimeric ΔBC RePC confirmed that the BCCP domain can adopt a wide range of catalytically productive motions. The subunits within the PC tetramer, however, exist in a more sterically constrained environment, necessitating a detailed evaluation of BCCP translocation in the context of the tetrameric enzyme. Furthermore, to evaluate the general applicability of these findings and to determine whether allosteric activation influences BCCP translocation, this phenomenon was studied in two different PC enzymes: a bacterial PC enzyme (*R. etli* PC; RePC) that is sensitive to acetyl CoA activation and a fungal PC enzyme (*A. nidulans* PC; AnPC) that is insensitive to acetyl CoA activation.

To individually assess the translocation pathways adopted by the BCCP domain during catalysis by both AnPC and RePC (Fig. 2), a series of hybrid PC tetramers were designed. These hybrids were comprised of two different mutated PCs (mutants 1 and 2) that had been inactivated in a combination of the catalytic and/or carrier domains (Fig. 4). We use the nomenclature ×BC, ×CT, and/or ×BCCP to indicate the inactivation of these domains in various constructs. Both the BC and CT domain were inactivated by mutations in the individual active sites, while the BCCP domain was rendered nonfunctional by mutating the

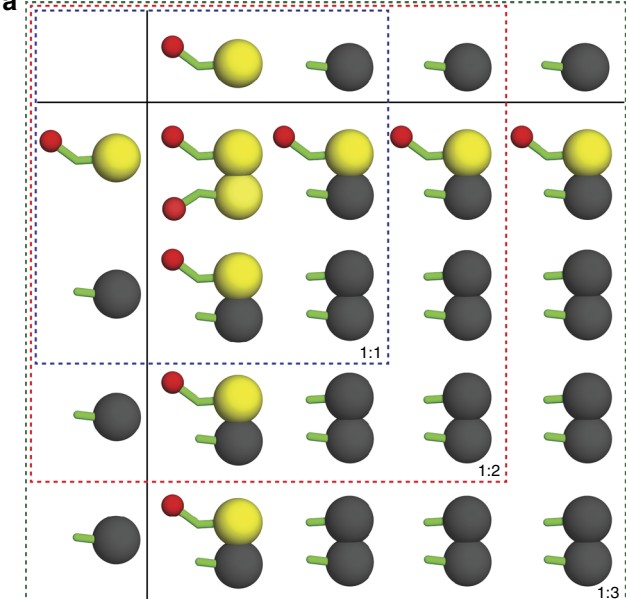

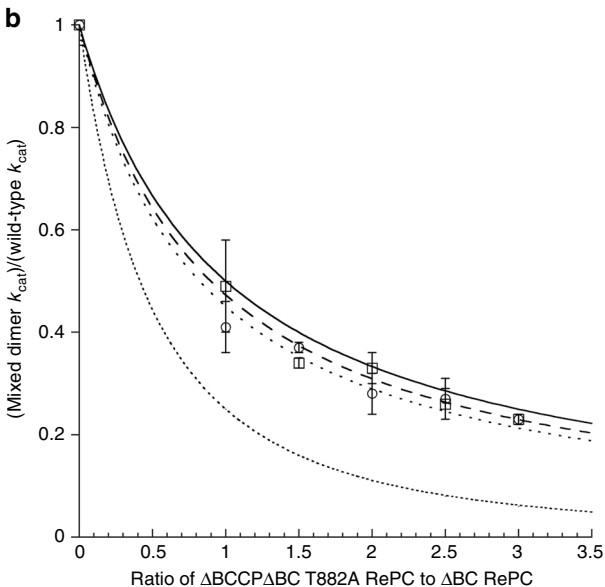

**Fig. 3** Intramolecular BCCP translocation. To assess intramolecular translocation, WT ΔBC RePC was diluted with increasing molar ratios of inactive ΔBCΔBCCP T882A RePC. **a** The predicted population of homodimers and heterodimers at different ratios of mixing WT ΔBC RePC with ΔBCΔBCCCP T882A RePC. **b** Two theoretical curves are predicted, depending on whether the residual activity of the mixed tetramers results 100% (solid line; Eq. (1)) or 0% (dotted line; Eq. (2)) from an intramolecular translocation of the BCCP domain. The experimental data, in the presence (dashed line, open squares) and absence (dotted line, open circles) of acetyl CoA was fit to an interpolation of these two equations (Eq. (3)) revealing ~80% intramolecular translocation of BCCP both in the presence and absence of acetyl CoA. Each data point represents the average of three independent measurements from one mixed population. Error bars represent the standard deviation

specific biotinylated lysine in the conserved MKME motif (Supplementary Table 1). The individual subunits expressed by these mutants can be represented as $M_1$ and $M_2$, respectively. In our approach, when $M_1$ and $M_2$ are co-expressed together, a heterogeneous population of homotetramers and hybrid

tetramers will form (Supplementary Fig. 2). Assuming that catalysis takes place only between subunits located on the same layer of the tetramer, all $M_1M_1$ or $M_2M_2$ combinations will remain inactive. When $M_1$ and $M_2$ combine on the same layer, however, activity will be recovered only if the functional BCCP domain is capable of a catalytically productive translocation between the two remaining functional active sites (Fig. 4).

When expressed and purified individually, the catalytic activity of each mutated PC enzyme was ~0, even in the presence of the allosteric activator, acetyl CoA (Tables 2 and 3). Thus, any catalytic activity recovered from a mixture of these mutated PC enzymes can only be the result of forming catalytically functional hybrid tetramers. Hybrids A, B, and C represent the catalytically productive enzymes corresponding to the translocation pathways a, b and c (Fig. 4). These hybrid tetramers can be used to unambiguously test three pathways (pathways a, b, and c in Fig. 2) of BCCP translocation during catalytic turnover.

Initially, hybrid tetramers were generated by mixing together two individually purified enzymes, as performed in several prior studies[2–4]. While this approach resulted in some recovered catalytic activity, the data suggested a low degree of hybridization between the two populations. In comparison, co-expression of two PC genes resulted in a significantly higher recovery of activity, consistent with a much higher degree of hybridization (Supplementary Table 2).

To unambiguously confirm that co-expression of two PC genes successfully generated a set of recombined hybrid tetramers, untagged wild-type RePC and fully inactivated N-terminally (His)$_9$-tagged RePC (RePC ×BC×CT×BCCP) were co-expressed using a pETDuet-1 vector in *Escherichia coli* BL21(DE3). In this system, wild-type RePC can only be co-purified if it recombines with (His)$_9$-tagged RePC ×BC×CT×BCCP to form hybrid tetramers. The degree of recombination into hybrid tetramers was assessed using an avidin gel shift assay. Successful recombination was expected to result in a purified population containing both untagged wild-type RePC (biotinylated) and (His)$_9$-tagged ×BC×CT×BCCP (unbiotinylated). A mobility shift in the presence of avidin confirmed the co-purification of wild type, untagged, biotinylated RePC, and demonstrated that hybrid tetramers were successfully generated by co-expression (Fig. 5). Using this same approach, the degree of biotinylation was assessed for all three co-expression products of RePC, confirming that ~50% of the protein population was biotinylated, as would be expected for a well-combined population of hybrid tetramers (Fig. 5).

A similar assessment was performed on AnPC, demonstrating the successful production of hybrid tetramers by co-expressing untagged wild-type AnPC with (His)$_9$-tagged AnPC ×BC×CT×BCCP. For reasons that are not clear, co-expression of mutated AnPC genes in the pETDuet-1 vector, in some cases, resulted in very poor expression levels for one of the two genes. This issue was resolved by converting to a co-expression method that involved co-transforming with two compatible plasmids, with each plasmid carrying an individually mutated PC gene. This approach successfully generated well-combined hybrid tetramers of AnPC (Fig. 5).

**PC adopts multiple carrier domain translocation pathways.** The steady-state catalytic activity of hybrid tetramers was measured with saturating concentration of substrates, in order to determine which translocation pathways are functional during catalysis. The system is designed such that, while all translocation pathways are available to any particular hybrid enzyme, only one pathway is catalytically productive. For example, the carrier domain of hybrid A may proceed through all four translocation

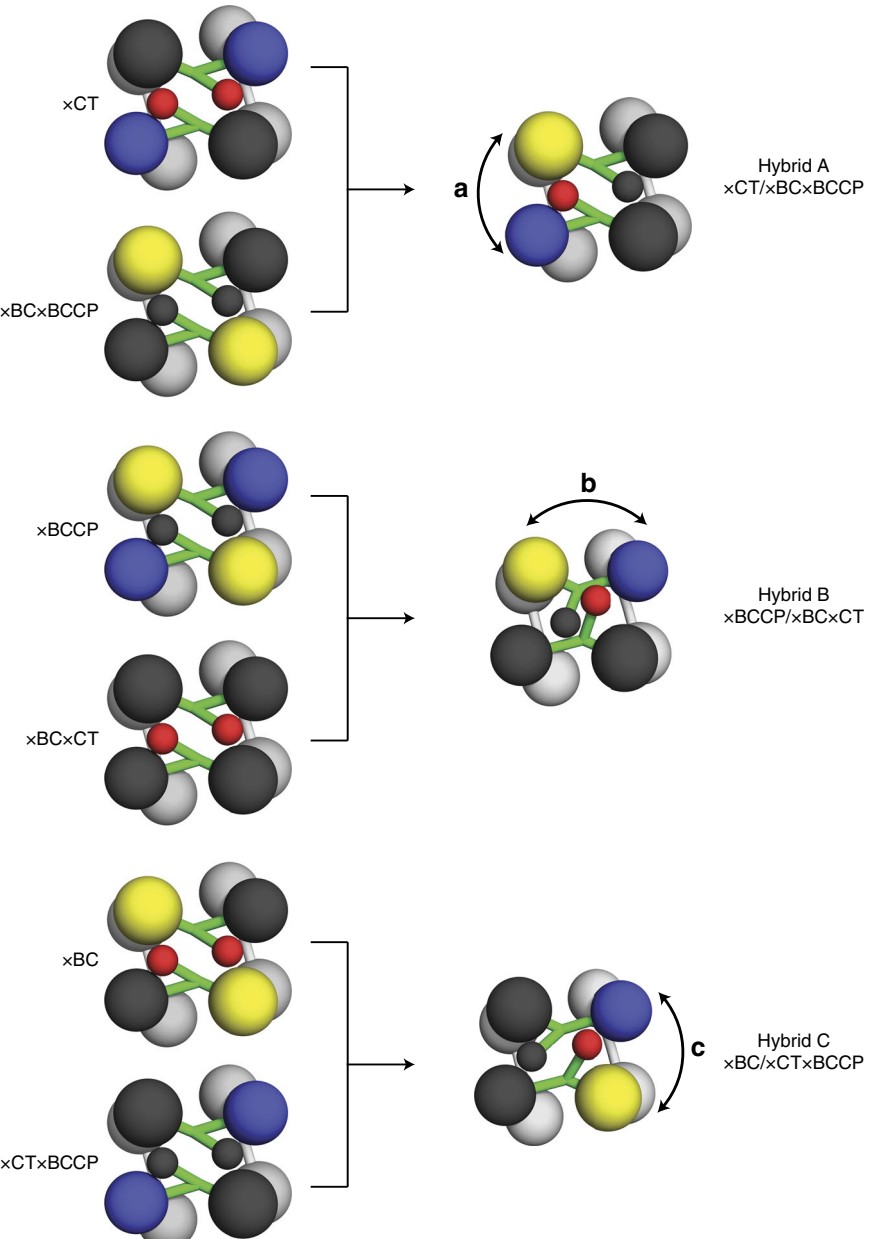

**Fig. 4** Design of hybrid tetramers to probe BCCP translocation. A schematic of the three hybrid tetramers (A-C) designed to probe the three specific intermolecular BCCP domain translocation pathways (**a**–**c**) in PC

pathways represented in Fig. 2, but only pathway a will be catalytically productive. Thus, while a hybrid system may adopt multiple carrier domain motions, it will only report on a single translocation pathway.

For both RePC and AnPC, the activities recovered in the hybrid tetramers were much higher than the activities of the individual PC mutants (Tables 2 and 3). This surprising result demonstrates that all three translocation pathways are catalytically functional in PC. Thus, both in the truncated dimer and the full-length tetramer, the BCCP domain is capable of using multiple catalytically productive translocation pathways.

In comparing the catalytic activities between different hybrid tetramers, it is tempting to draw conclusions about the relative contribution of one translocation pathway over another. For example, the activities measured from the hybrid tetramers in AnPC (Table 3) vary, potentially indicating that translocation

pathway a contributes more to the catalytic turnover than b and c in AnPC. Theoretically, if each mixed population resulted in an equal degree of recombination between the two mutants $M_1$ and $M_2$ (Supplementary Fig. 2), the activities could serve as a useful measure of the relative catalytic turnover derived from each translocation pathway. However, it must be noted that the total degree of recombination during co-expression could not be fully controlled or determined. Thus, while these results clearly establish that each pathway does contribute to catalysis, the current approach cannot determine the relative contribution that each pathway makes to catalytic turnover in the wild-type enzyme.

**Acetyl CoA activates a single translocation pathway**. Acetyl CoA significantly enhances the coupling efficiency between the biotin carboxylation reaction in the BC domain and the carboxyl

**Table 2 Activation by acetyl coenzyme A of wild type and mutated RePC and hybrid tetramers**

| RePC mutants | [a] $k_{cat}^{AcoA}$ (min$^{-1}$) | [b] RePC hybrid tetramers | $k_{cat}^{AcoA}$ (min$^{-1}$) | $k_{cat}^0$ (min$^{-1}$) | $k_{cat}^{AcoA}$/$k_{cat}^0$ | $K_a$ (μM) | h | [c] BCCP Translocation pathways |
|---|---|---|---|---|---|---|---|---|
| RePC wild type | - | - | 1450 ± 30 | 130 ± 10 | 11 ± 1 | 3.2 ± 0.1 | 4.5 ± 0.2 | |
| ×CT | 0.05 ± 0.01 | | | | | | | |
| ×BC×BCCP | 0.8 ± 0.3 | ×CT/×BC×BCCP | 320 ± 4 | 12 ± 4 | 27 ± 9 | 5.7 ± 0.1 | 5.6 ± 0.7 | |
| ×BC | 0.17 ± 0.05 | | | | | | | |
| ×CT×BCCP | 1.7 ± 0.4 | ×BC/×CT×BCCP | 12 ± 1 | 13 ± 1 | 1.0 ± 0.1 | ND | ND | |
| ×BCCP | 0.16 ± 0.25 | ×BCCP/×BC×CT | 10 ± 1 | 9 ± 1 | 1.1 ± 0.2 | ND | ND | |
| ×BC×CT | 0.21 ± 0.12 | | | | | | | |

[a]All $k_{cat}$ values were determined using the subunit molecular weight of RePC (126 kDa). Each subunit contributes a single BC–CT active site pair. The $k_{cat}^{AcoA}$ for pyruvate carboxylation by individual domain mutants of RePC was measured in the presence of 25 mM NaHCO$_3$, 2.5 mM MgATP, 12 mM pyruvate and 0.25 mM acetyl CoA. The reported values are the average of three independent measurements from one purified sample. Errors are reported as the standard deviation
[b]The kinetic constants $k_{cat}^{AcoA}$ and $k_{cat}^0$ for the RePC hybrid tetramers are reported as the average of four independent measurements from one purified sample. Errors are reported as the standard deviation. The kinetic constants $K_a$ and h are derived from the curve fit of the data from one purified sample, where each independent acetyl CoA concentration was measured four independent times (Supplementary Fig. 3). The standard errors are reported for $K_a$ and h. All assays were performed in the presence of 25 mM NaHCO$_3$, 2.5 mM MgATP, 12 mM pyruvate and 0–0.5 mM acetyl CoA
[c]Black shading indicates an inactive domain; light shading indicates an active domain. Arrows represent a catalytically productive translocation

transfer reaction in the CT domain of PC[12]. This enhanced-coupling efficiency may result from acetyl CoA regulating the BCCP translocation pathway in PC, but this has never been systematically investigated.

In order to determine whether the enhanced-coupling efficiency of RePC in the presence of acetyl CoA occurs through the regulation of BCCP domain translocation, the three populations of RePC hybrid tetramers were kinetically characterized in the presence of acetyl CoA (Table 2). The $K_a$ value determined for acetyl CoA in hybrid A was very similar to that determined for wild-type RePC, indicating that acetyl CoA binding is not disrupted in this hybrid tetramer. The overall degree of acetyl CoA activation is significantly higher in hybrid A than it is in the wild-type enzyme (Table 2; ~30-fold vs ~10-fold). Most notably, only RePC hybrid A was significantly activated by acetyl CoA, while the two other hybrid tetramers were not. Thus, acetyl CoA specifically activates pathway a in RePC, while it has no effect on pathways b and c.

While RePC and AnPC both enable multiple carrier domain translocation pathways, AnPC is insensitive to acetyl CoA activation. To determine whether the activation of translocation pathway a by acetyl CoA correlates with acetyl CoA sensitivity, similar experiments were performed in AnPC. Acetyl CoA did not activate any BCCP domain translocation pathway in AnPC (Table 3). Thus, the activation of translocation pathway a is dependent on the sensitivity of the enzyme to allosteric activation by acetyl CoA.

**Aspartate inhibition is independent of carrier translocation**. L-aspartate is a well-characterized allosteric inhibitor of PC. Studies in both RePC and AnPC have suggested that the binding sites for

L-aspartate and acetyl CoA are distinct, but that the binding of these two contrasting allosteric effectors is mutually exclusive[8,9].

To investigate whether L-aspartate inhibition in RePC also results in changes to BCCP domain translocation, L-aspartate inhibition was measured for each of the three hybrid tetramers in the absence of acetyl CoA (Table 4). In contrast to what was observed with acetyl CoA activation, all three of the hybrid tetramers were inhibited by L-aspartate. In the absence of acetyl CoA, the $K_i$ values determined for L-aspartate with each of the hybrid tetramers were very similar to the $K_i$ value determined for the wild-type enzyme, indicating that L-aspartate binding was not disrupted in the hybrid tetramers. Whereas RePC hybrid tetramer A was specifically activated by acetyl CoA, its inhibition by L-aspartate was no greater than inhibition in any other RePC hybrid tetramer. When this experiment was repeated in the presence of acetyl CoA, wild-type RePC and all of the RePC hybrid tetramers were again inhibited by L-aspartate (Table 4). For both wild-type RePC and the three hybrid tetramers, the $K_i$ values for L-aspartate determined in the presence of acetyl CoA were significantly increased as a result of the competition for binding with acetyl CoA.

As expected, the degree of inhibition in the presence of acetyl CoA was enhanced for wild-type RePC due to acetyl CoA activation. Interestingly, the degree of hybrid tetramer inhibition by L-aspartate in the presence of acetyl CoA is exaggerated only for RePC hybrid A. This further confirms that only RePC hybrid tetramer A is sensitive to acetyl CoA activation.

L-aspartate inhibition was measured for each of the three hybrid tetramers of AnPC in both the presence and absence of acetyl CoA (Table 5). The inhibition of AnPC by L-aspartate revealed that, as with RePC, the $K_i$ values were increased in the presence of acetyl CoA, suggesting that acetyl CoA and L-

**Table 3 Activation by acetyl coenzyme A of wild type and mutated AnPC and hybrid tetramers**

| AnPC mutants | [a] $k_{cat}^{AcoA}$ (min$^{-1}$) | [b] AnPC hybrid tetramers | $k_{cat}^{AcoA}$ (min$^{-1}$) | $k_{cat}^{0}$ (min$^{-1}$) | $k_{cat}^{AcoA}$ /$k_{cat}^{0}$ | $K_a$ (µM) | h | [c] BCCP Translocation pathways |
|---|---|---|---|---|---|---|---|---|
| AnPC wild type | - | - | 1910 ± 100 | 2030 ± 110 | 0.9 ± 0.1 | ND | ND |  |
| ×CT | 0.39 ± 0.07 | ×CT/×BC×BCCP | 90 ± 10 | 102 ± 6 | 0.9 ± 0.1 | ND | ND |  |
| ×BC×BCCP | 3.1 ± 0.4 | | | | | | | |
| ×BC | 0.16 ± 0.08 | ×BC/×CT×BCCP | 30 ± 1 | 32 ± 1 | 0.9 ± 0.1 | ND | ND |  |
| ×CT×BCCP | 0.33 ± 0.04 | | | | | | | |
| ×BCCP | 1.8 ± 0.7 | ×BCCP/×BC×CT | 11 ± 1 | 10 ± 1 | 1.1 ± 0.1 | ND | ND |  |
| ×BC×CT | 0.99 ± 0.26 | | | | | | | |

[a] $k_{cat}$ values were determined using the subunit molecular weight of AnPC (130 kDa). Each subunit contributes a single BC–CT active site pair. $k_{cat}^{AcoA}$ for pyruvate carboxylation by individual domain mutants of AnPC was measured in the presence of 25 mM NaHCO$_3$, 2.5 mM MgATP, 12 mM pyruvate and 0.1 mM acetyl CoA. The reported values are the average of three independent measurements from one purified sample. Errors are reported as the standard deviation
[b] The kinetic constants $k_{cat}^{AcoA}$ and $k_{cat}^{0}$ for the AnPC hybrid tetramers are reported as the average of four independent measurements from one purified sample. Errors are reported as the standard deviation. All assays were performed in the presence of 25 mM NaHCO$_3$, 2.5 mM MgATP, 12 mM pyruvate and 0 or 0.5 mM acetyl CoA
[c] Black shading indicates an inactive domain; light shading indicates an active domain. Arrows represent a catalytically productive translocation

aspartate compete for binding to AnPC. This confirms that acetyl CoA does, indeed, bind to AnPC, even though it does not activate the overall rate of the AnPC-catalyzed reaction. Since AnPC is insensitive to acetyl CoA activation, the degree of inhibition from L-aspartate was not enhanced in the presence of acetyl CoA for any of the AnPC hybrid tetramers.

To summarize, all three of the hybrid tetramers in both RePC and AnPC are inhibited by L-aspartate in the absence of acetyl CoA. In the presence of acetyl CoA, only hybrid tetramer A from RePC is sensitive to enhanced inhibition by L-aspartate, consistent with the unique sensitivity of translocation pathway a to acetyl CoA activation.

## Discussion

This study sought to unambiguously assess the productivity of three intermolecular BCCP translocation pathways in catalytically distinct PC enzymes from two different species. Contrary to the accepted dogma, which states that the BCCP domain transits exclusively between a single combination of BC and CT domains, the present study reveals a much more fluid and flexible carrier domain that travels in catalytically productive pathways between multiple combinations of active site pairs. Considering that RePC and AnPC are evolutionarily and functionally divergent, and that the kinetic and thermodynamic features of the vast majority of PC enzymes are highly conserved, we believe that these findings are broadly applicable to all PC enzymes. Notably, a recently reported structure of PC from *Lactococcus lactis* revealed the BCCP domain uniquely posed in an interaction with the BC domain on a neighboring subunit (equivalent to the position occupied through translocation pathways b and c)[18]. The present study confirms and validates the catalytic relevance of this structural snapshot and extends the implications to a broad cross-section of PC enzymes.

The three hybrid tetramers assessed in this study were designed to examine three possible intermolecular translocation pathways (Fig. 2, pathways a, b, and c) by measuring recovered catalytic activity from the products of co-expressed PC domain-inactivated mutations. The fourth translocation (Fig. 2, pathway d) involves an intramolecular pathway and, therefore, cannot be easily assessed. In principle, intramolecular pathway d can be isolated and evaluated using hybrid tetramers generated between a wild-type PC and a triple domain-inactivated mutant (×BC×CT×BCCP); this combination ensures that, for the hybrid tetramer, pathway d is the only catalytically viable pathway. However, co-expression results in a heterogeneous population that includes a significant proportion of catalytically active wild-type homotetramer. Since neither mixing nor co-expression techniques allow for a predictable, fully heterogeneous population of tetramers, the catalytic contribution from wild-type homo-tetramers and hybrid tetramers cannot be disentangled, precluding an accurate assessment of translocation pathway d in the tetramer. Nevertheless, the flexibility observed in the dimeric ΔBC RePC (Fig. 3), indicates that the BCCP domain is fully capable of adopting an intramolecular interaction with the CT domain of the same subunit. Thus, translocation pathway d is also very likely to contribute to catalytic turnover in the PC tetramer.

Intermediate transfer via carrier domain translocation is essential to ensure a high-coupling efficiency between the individual half-reactions in PC[12,13]. Several studies have focused on specific mechanistic features in both the BC and CT domains that contribute to the enzyme's high-coupling efficiency[3,4,12,19]. These

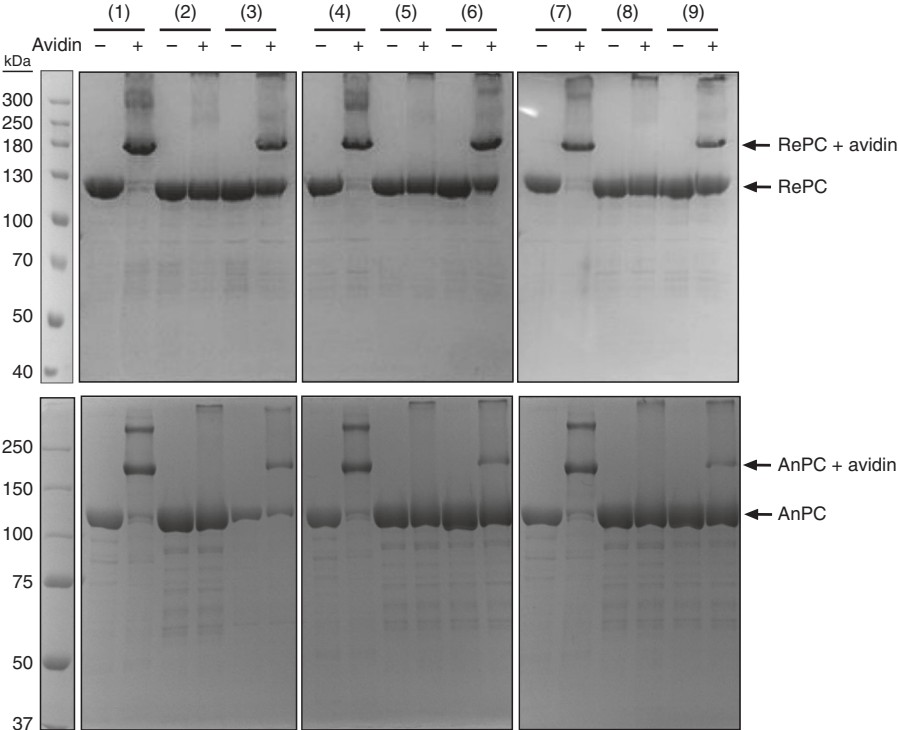

**Fig. 5** Hybrid tetramers are produced by co-expression. Avidin gel shift assays showing the successful production of all three co-expression products of RePC and AnPC. (1) ×BC; (2) ×CT×BCCP; (3) ×BC/×CT×BCCP; (4) ×CT; (5) ×BC×BCCP; (6) ×CT/×BC×BCCP; (7) ×BC×CT; (8) ×BCCP; (9) ×BCCP/×BC×CT

studies point to specific gatekeeping features in the individual active sites that reduce the potential for abortive ATP cleavage and that ensure a high probability of carboxyl-group transfer from carboxybiotin to pyruvate in the CT domain. Interestingly, in the absence of acetyl CoA, the efficiency with which MgATP cleavage at the BC domain is coupled with oxaloacetate formation at the CT domain is severely reduced, even at saturating concentrations of all substrates[12]. It is known that acetyl CoA accelerates MgATP cleavage in the BC domain[10], but the enhanced-coupling efficiency in the presence of acetyl CoA indicates that there are additional factors, beyond substrate-mediated control of the individual active sites, which contribute to efficient intermediate channeling in PC. Since intermediate transfer via carrier domain translocation is essential to link the catalytic activities of the individual half-reactions, we reasoned that acetyl CoA might enhance the coupling efficiency by mediating BCCP translocation in PC. The intramolecular translocation of BCCP in the dimeric enzyme form of RePC (ΔBC RePC) was not sensitive to acetyl CoA in the oxamate-induced oxaloacetate decarboxylation reaction (Fig. 3b). However, the BC domain truncation eliminates the portion of the acetyl CoA binding site which resides at the BC domain dimer interface and, consequently, acetyl CoA has been shown not to alter the catalytic activity of ΔBC RePC[2,5,16]. Thus, the impact of acetyl CoA on BCCP translocation can only be evaluated in the context of the intact tetramer. Recent thermodynamic and kinetic studies have shown that acetyl CoA facilitates the long-range communication between the individual BC and CT domains in the intact PC tetramer and have suggested that the BCCP carrier domain may play a role in mediating this communication[3,13]. For the acetyl CoA sensitive RePC, our results offer direct evidence that acetyl CoA activates a single, specific translocation pathway: the translocation of BCCP from the BC domain of its own subunit to the

CT domain of the neighboring subunit (pathway a, Fig. 2). This offers a compelling mechanistic explanation for the enhanced-coupling efficiency that has been observed in the presence of acetyl CoA[12,13]. While the molecular mechanism by which acetyl CoA promotes one translocation pathway over the others is not currently clear, it may be related to subtle structural rearrangements in the individual domains, such as those that have been reported for *S. aureus* PC in the presence of acetyl CoA (5).

Prior studies have reported detailed thermodynamic analyses of PC catalysis, where acetyl CoA has been shown to decrease the activation entropy for the reaction catalyzed by both chicken liver[20] and *S. aureus* PC[13], indicating that acetyl CoA facilitates a more ordered thermodynamic state during catalytic turnover. This reduction in activation entropy was originally attributed to more compact quaternary structures that were observed in low-resolution cryo-EM images in the presence of acetyl CoA[21]. However, several more recent high-resolution X-ray crystal and cryo-EM structures in the presence and absence of acetyl CoA have not provided any corroborating evidence to support a more compact PC quaternary structure in the presence of acetyl CoA[1,2,4,5]. Furthermore, the acetyl CoA insensitive AnPC was also shown in cryo-EM studies to adopt a more compact structure in the presence of acetyl CoA[22]. This leads us to believe that the compact structure observed in the presence of acetyl CoA is more likely to be a general structural feature of all PC enzymes and that it is not related to the observed reduction in the activation entropy. Rather, the decreased activation entropy in the presence of acetyl CoA is more easily rationalized by the constrained freedom of the BCCP domain during catalytic turnover. Taken together, we propose that, for acetyl CoA sensitive PC enzymes like RePC, the BCCP domain is flexible in the absence of acetyl CoA and is able to adopt at least four separate catalytically productive translocation pathways. In the presence of acetyl CoA, a

**Table 4 Inhibition of wild-type RePC and RePC hybrid tetramers by L-aspartate in the presence and absence of acetyl coenzyme A**

| RePC constructs | + 0 mM acetyl coenzyme A | | | | | + 0.25 mM acetyl coenzyme A | | | | |
|---|---|---|---|---|---|---|---|---|---|---|
| | $^a k_{cat}^0$ (min$^{-1}$) | $k_{cat}^{asp}$ (min$^{-1}$) | $k_{cat}^{asp}/k_{cat}^0$ | $K_i$ (mM) | $h$ | $k_{cat}^0$ (min$^{-1}$) | $k_{cat}^{asp}$ (min$^{-1}$) | $k_{cat}^{asp}/k_{cat}^0$ | $K_i$ (mM) | $h$ |
| Wild type | 150 ± 8 | 81 ± 2 | 0.54 ± 0.03 | 4.7 ± 0.7 | 1.2 ± 0.2 | 1200 ± 60 | 180 ± 40 | 0.15 ± 0.03 | 138 ± 5 | 4.8 ± 0.7 |
| RePC_×CT/×BC×BCCP | 11.5 ± 0.8 | 7.4 ± 0.3 | 0.64 ± 0.05 | 6.2 ± 0.5 | 1.6 ± 0.1 | 240 ± 7 | 34 ± 2 | 0.14 ± 0.01 | 26 ± 1 | 1.6 ± 0.1 |
| RePC_×BC/×CT×BCCP | 10 ± 1 | 2.8 ± 0.4 | 0.28 ± 0.05 | 8.2 ± 1.3 | 1.3 ± 0.2 | 12.5 ± 0.4 | 2.9 ± 0.1 | 0.23 ± 0.01 | 61 ± 6 | 1.2 ± 0.1 |
| RePC_×BCCP/×BC×CT | 7.9 ± 0.2 | 2.3 ± 0.3 | 0.29 ± 0.04 | 9.7 ± 1.5 | 1.2 ± 0.2 | 10.5 ± 0.8 | 3.3 ± 0.3 | 0.31 ± 0.04 | 53 ± 5 | 1.2 ± 0.1 |

$^a k_{cat}$ values were determined using the subunit molecular weight of RePC (126 kDa). Each subunit contributes a single BC–CT active site pair. The kinetic constants $k_{cat}^0$ and $k_{cat}^{asp}$ are reported as the average of four independent measurements from one purified sample. Errors are reported as the standard deviation. The kinetic constants $K_i$ and $h$ are derived from the curve fit of the data from one purified sample, where each independent L-aspartate concentration was measured four independent times (Supplementary Fig. 5). The standard errors are reported for $K_i$ and $h$. All assays were performed in the presence of 25 mM NaHCO$_3$, 2.5 mM MgATP, 12 mM pyruvate, ±0.25 mM acetyl CoA and 0–300 mM L-aspartate

**Table 5 Inhibition of wild-type AnPC and AnPC hybrid tetramers by L-aspartate in the presence and absence of acetyl coenzyme A**

| AnPC constructs | + 0 mM acetyl coenzyme A | | | | | + 0.1 mM acetyl coenzyme A | | | | |
|---|---|---|---|---|---|---|---|---|---|---|
| | $^a k_{cat}^0$ (min$^{-1}$) | $k_{cat}^{asp}$ (min$^{-1}$) | $k_{cat}^{asp}/k_{cat}^0$ | $K_i$ (mM) | $h$ | $k_{cat}^0$ (min$^{-1}$) | $k_{cat}^{asp}$ (min$^{-1}$) | $k_{cat}^{asp}/k_{cat}^0$ | $K_i$ (mM) | $h$ |
| Wild type | 2200 ± 90 | 108 ± 8 | 0.049 ± 0.004 | 2.6 ± 0.1 | 1.7 ± 0.1 | 1970 ± 100 | 123 ± 7 | 0.062 ± 0.005 | 3.8 ± 0.3 | 2.7 ± 0.4 |
| AnPC_×CT/×BC×BCCP | 84 ± 3 | 3.1 ± 0.5 | 0.037 ± 0.006 | 1.8 ± 0.2 | 1.7 ± 0.4 | 94 ± 8 | 4.5 ± 0.2 | 0.048 ± 0.005 | 7.5 ± 0.2 | 3.1 ± 0.3 |
| AnPC_×BC/×CT×BCCP | 31.4 ± 1.3 | 2.6 ± 0.3 | 0.08 ± 0.01 | 2.3 ± 0.1 | 1.8 ± 0.2 | 34 ± 3 | 3.4 ± 0.4 | 0.10 ± 0.01 | 13 ± 1 | 2.0 ± 0.2 |
| AnPC_×BCCP/×BC×CT | 10 ± 1 | 1.1 ± 0.2 | 0.11 ± 0.21 | 1.86 ± 0.02 | 2.1 ± 0.1 | 9.4 ± 0.5 | 1.1 ± 0.1 | 0.12 ± 0.01 | 10 ± 1 | 1.7 ± 0.2 |

$^a k_{cat}$ values were determined using the subunit molecular weight of AnPC (130 kDa). Each subunit contributes a single BC–CT active site pair. The kinetic constants $k_{cat}^0$ and $k_{cat}^{asp}$ are reported as the average of four independent measurements from one purified sample. Errors are reported as the standard deviation. The kinetic constants $K_i$ and $h$ are derived from the curve fit of the data from one purified sample, where each independent L-aspartate concentration was measured four independent times (Supplementary Fig. 6). The standard errors are reported for $K_i$ and $h$. All assays were performed in the presence of 25 mM NaHCO$_3$, 2.5 mM MgATP, 12 mM pyruvate, ±0.1 mM acetyl CoA and 0–100 mM L-aspartate

single translocation pathway (pathway a, Fig. 2) dominates, accounting for the decreased activation entropy observed in the presence of acetyl CoA.

The increased sensitivity of RePC Hybrid A to activation by acetyl CoA (Table 2; ~30-fold rate enhancement) compared to wild-type RePC (Table 2; ~10-fold rate enhancement) is fully consistent with the above hypothesis. In Hybrid A, only one of the four translocation pathways is catalytically productive, resulting in a reduced baseline activity compared to wild-type RePC, where all four translocation pathways yield productive turnover. However, in the presence of acetyl CoA, the BCCP domain predominantly transits through translocation pathway a in both wild-type RePC and Hybrid A. This enhances the effect of acetyl CoA on the catalytic activity of Hybrid A and contributes to its increased allosteric sensitivity.

Unlike acetyl CoA activation, which enhances the coupling efficiency between the two half-reactions in PC, L-aspartate does not alter the enzyme coupling efficiency[8]. In both RePC and AnPC, all three translocation pathways were inhibited by L-aspartate (Tables 4 and 5), indicating that the allosteric inhibition by L-aspartate is not mediated through effects on BCCP translocation. It is clear from the present study that acetyl CoA and L-aspartate adopt different mechanisms in regulating PC activity. The competing effects of acetyl CoA and L-aspartate on PC catalysis are, therefore, most easily explained by their mutually exclusive binding[8,9].

Carrier proteins and carrier domains are commonly used to shuttle covalently bound intermediates between remote active sites in large, multienzyme complexes, such as fatty acid synthases (FAS)[23], polyketide synthases (PKS)[24], non-ribosomal peptide synthetases (NRPS)[25], pyruvate dehydrogenase[26], and the broader family of biotin-dependent enzymes[27]. In all cases, the transfer of reaction intermediates between spatially distinct active sites requires long-range movements of the carrier domain during catalytic turnover. In NRPS, for example, the peptidyl carrier protein domain rotates ~75° and translocates ~60 Å as it transports intermediates between active sites[25]. Such long-range motions between multiple active sites require the carrier domain to combine remarkable flexibility with exquisite selectivity in order to appropriately present the proper intermediates to the correct active sites. For type I modular PKS and NRPS, the carrier domain transfers reaction intermediates among multiple modules and mediates specificity through protein-protein interactions between the carrier domain and the individual catalytic domains[28,29]. Iterative enzyme complexes, such as yeast type I FAS, use separate reaction chambers to constrain and control intermediate transfer and avoid off-pathway reactions with improperly positioned intermediates[30]. Consistent with the model established for FAS, it has long been envisioned that PC confines the carrier domain to a single reaction chamber between one set of active site pairs. Unlike most assembly line enzymes, however, PC is not at risk for off-pathway reactions when the carrier domain travels between alternate combinations of active site pairs. Instead, PC presents a new example for catalytic control in a swinging domain enzyme, through regulated movement of the carrier domain in the presence of the allosteric activator, acetyl CoA. Instead of constraining carrier movements as a means to control specificity, altered carrier domain movements in PC contribute to enhanced catalytic efficiency. These studies reveal a new mechanism for carrier domain movement and allosteric regulation in swinging domain enzymes.

## Methods

**Cloning, expression, and purification of truncated RePC.** The original ΔBC RePC and ΔBCΔBCCP RePC constructs in pET-28a[4,19] were re-cloned into compatible expression vectors. To ensure vector compatibility, the pET-28a vector encoding ΔBCΔBCCP RePC was modified to replace the pBR322 origin of replication with a pSC101 origin of replication. The gene encoding ΔBCΔBCCP RePC was re-cloned into vector pKLD66nCBP[31]. The T882A mutation was introduced into both ΔBC RePC and ΔBCΔBCCP RePC by whole plasmid mutagenesis using the Quikchange method. The N-terminally poly(His)-tagged ΔBCΔBCCP RePC was expressed in *E. coli* BL21Star(DE3) cells from a modified pET-28a-(His)$_8$-TEV vector[4]. ΔBC RePC with an N-terminal CaM binding domain was expressed from vector pKLD66nCBP and co-expressed with BirA encoded by the pCY216 vector (4). Proteins were produced using a 20 L batch culture in M9 minimal medium (containing 200 mg/L ampicillin, 30 mg/L chloramphenicol, and 1 mg/L biotin). After reaching an optical density (600 nm) of 1.0, the culture was induced for 24 h at 16 °C with 1 mM isopropyl 1-thio-β-D-galacto-pyranoside (IPTG) and 20 mM L-arabinose[4]. The cell paste from the equivalent of 6 L of M9 minimal media was resuspended in 150 mL of lysis buffer (20 mM Tris-HCl, pH 7.8; 2 mM CaCl$_2$; 200 mM NaCl; 6 mM β-mercaptoethanol; 5 μM epoxysuccinyl-L-leucylamido (4-guanida) butane (E-64); 1 μM pepstatin A; 1 mM phenylmethylsulfonyl fluoride; and 200 μg/mL lysozyme). The cell suspension was lysed by sonication and pelleted by centrifugation. The supernatant was loaded on a 10 mL column of CaM resin at a flow rate of 1.5 mL per minute. The column was subsequently washed with 300 mL of lysis buffer at a flow rate of 1.5 mL per minute. Following the wash, the protein was eluted with 50 mL of elution buffer (20 mM Tris-HCl, pH 7.8; 2 mM ethylene glycol-bis(β-aminoethyl ether)-N,N,N′,N′-tetraacetic acid (EGTA); 200 mM NaCl; 6 mM β-mercaptoethanol) at a flow rate of 1.5 mL per minute. Fractions containing the protein were pooled and dialyzed (20 mM Tris-HCl, pH 7.8; 200 mM NaCl; 6 mM β-mercaptoethanol) for 13 h. The protein was concentrated to 7.3 mg/mL (ΔBC RePC) and 1.5 mg/mL (ΔBC T882A RePC) as determined by the protein's predicted extinction coefficient and absorbance at 280 nm. The concentrated protein was drop frozen in liquid nitrogen and stored at −80 °C.

**Generation of the RePC and AnPC domain-inactivated mutants.** The RePC and AnPC wild-type genes in a pET-17b-(His)$_9$ vector were site-specifically mutated by whole plasmid mutagenesis using the Quikchange method. The specific mutations used to inactivate individual domains of both RePC and AnPC are provided in Supplementary Table 2. The correct gene sequence for each set of mutations was confirmed by whole gene sequencing by Functional Biosciences (Madison, WI).

**Cloning into compatible co-expression vectors.** RePC hybrid tetramers were generated by co-expressing two individual domain mutants on a pETDuet-1 vector. Pairs of genes encoding poly(His)$_9$-tagged RePC domain inactive mutants were inserted into the pETDuet-1 vector in tandem by PCR-based amplification from pET-17b followed by ligation-based cloning into the pETDuet-1 vector. In all cases, one of the domain-inactivated RePC genes was inserted into the first multiple cloning site (MCS) using the XbaI/NotI restriction endonuclease sites; the second gene was inserted into the second MCS using the NdeI/PacI restriction endonuclease sites (Supplementary Table 3). In all cases, both domain-inactivated mutant genes encode a recombinant poly(His)$_9$-tag at the protein N-terminus.

The co-expression of pairs of AnPC mutants for the generation of hybrid tetramers was performed by the co-transformation of two compatible plasmids, pSFDuet-1 and pET-17b, with one member of the pair encoded on pSFDuet-1, while the other member of the pair was encoded on pET-17b (Supplementary Table 3). Three domain-inactivated mutants were maintained in pET-17b while three complementary domain-inactivated mutants were cloned into the pRSFDuet-1 MCS using the NcoI/PacI restriction endonuclease sites (Supplementary Table 3). In this system, the domain-inactivated mutant genes encoded on pET-17b included a recombinant poly(His)$_9$-tag at the protein N-terminus, while those genes encoded on pRSFDuet-1 did not include an affinity tag.

**Protein expression and purification of full-length PC.** All full-length PC genes were expressed and purified in the same manner. In order to ensure complete biotinylation, all PC clones were co-transformed and co-expressed with *E. coli* biotin protein ligase A (BirA) on vector pCY216 (ref. [32]). Transformed *E. coli* BL21 (DE3) cells were cultured in M9 minimal media at 37 °C to an optical density (600 nm) of 0.8 −0.9, whereupon cells were induced by adding IPTG and L-arabinose to a final concentration of 1 and 25 mM, respectively. In addition, the culture was supplemented with D-(+)-biotin and MnCl$_2$ to a final concentration of 3 mg/L and 0.1 mM, respectively. Induced cells were incubated with shaking at 16 °C for 20 h before harvesting by centrifugation.

All PC full-length enzymes were purified using sequential Ni$^{2+}$-affinity and anion exchange chromatography. The cell paste from 12 L of cell culture was resuspended in 200 mL Ni-lysis buffer (20 mM Tris-HCl pH 7.8; 200 mM NaCl; 0.5 mM EGTA; 5 mM imidazole; 6 mM β-mercaptoethanol; 1 mM PMSF; 1 μM pepstatin A; and 5 μM E-64). Cells were disrupted by sonication at a temperature not exceeding 10 °C and pelleted by centrifugation. The supernatant was loaded on a 10 mL Ni$^{2+}$-nitrilotriacetic acid profinity resin column (Bio-Rad, Hercules, CA).

The column was washed with 20× column volume of wash buffer buffer (20 mM Tris-HCl pH 7.8; 200 mM NaCl; 0.5 mM EGTA; 20 mM imidazole; 6 mM β-mercaptoethanol) and the protein was eluted with a gradient from 20 to 250 mM imidazole using wash buffer and elution buffer (20 mM Tris-HCl pH 7.8; 200 mM NaCl; 0.5 mM EGTA; 250 mM imidazole; 6 mM β-mercaptoethanol). Purified protein was pooled and dialyzed against a buffer compatible for anion exchange chromatography (20 mM triethanolamine, pH 8.0; 50 mM NaCl; 1 mM EGTA; and 2 mM dithiothreotol (DTT)) at 4 °C overnight. The dialyzed protein was loaded on a 10 mL Q-Sepharose Fast Flow resin column (GE Healthcare) and was eluted from the column in buffer (20 mM triethanolamine, pH 8.0; 50 mM NaCl; 1 mM EGTA; and 2 mM DTT) with a gradient from 50 mM to 1 M NaCl. The elution peak of PC full-length protein was typically between 400 and 800 mM NaCl. The purified protein was pooled and dialyzed against storage buffer (10 mM Tris-HCl pH 7.8; 50 mM NaCl; 10 mM MgCl$_2$, 5% (v/v) glycerol, and 2 mM DTT) for >4 h with at least two successive changes. The protein was concentrated to a final concentration of 6–10 mg/mL and was drop frozen in liquid nitrogen prior to storage at −80 °C freezer. All protein concentrations were determined by the predicted extinction coefficient and absorbance at 280 nm[33].

**Avidin gel shift assays.** Avidin gel shift assays were performed to measure the degree of PC biotinylation. Avidin is a homotetramer of molecular weight ~60 kDa with four very high-affinity biotin binding sites per avidin tetramer. Chicken egg white avidin (Sigma, 1.2 mg/mL) and denatured PC protein (1 mg/mL) were prepared in buffer (50 mM HEPES, pH 7.5, 200 mM NaCl) and mixed in equal volumes to a final molar ratio of 10:1, avidin:PC. The mixture was incubated at room temperature (25 °C) for 40 min. Biotinylated PC binds with avidin irreversibly, retarding its migration on SDS-PAGE. In contrast, unbiotinylated PC does not bind with the avidin and migrates on SDS-PAGE at a molecular weight of a single subunit, ~125 kDa.

**Enzyme assays.** Pyruvate carboxylation activity was monitored spectrophotometrically at 340 nm by following the conversion of oxaloacetate to malate using the coupled enzyme malate dehydrogenase (MDH). Reactions were performed in 0.1 M Tris-HCl (pH 7.8), 7 mM MgCl$_2$, and 0.1 M KCl. All substrates and coupling reagents were mixed in a 10× stock solution that containing components at the following concentrations: 250 mM NaHCO$_3$, 25 mM MgATP, 120 mM pyruvate, 2.4 mM NADH, and 100 units/mL malate dehydrogenase. For titration of acetyl CoA and L-aspartate, the final concentration of acetyl CoA ranged from 0 to 0.5 mM and L-aspartate ranged from 0 to 300 mM. All reactions were performed in 96-well plate format at 25 °C in a total working volume of 200 μL. The final PC concentration in the assay ranged between 5 and 50 μg/mL (0.04–0.4 μM) per reaction. Reactions were initiated by addition of 20 μL of the 10× substrate stock solution.

The PC-catalyzed rate of oxaloacetate decarboxylation was determined by measuring the reduction of pyruvate to lactate using lactate dehydrogenase[16]. Unless otherwise noted, reactions were performed in 0.1 M Tris-HCl (pH 7.8), 0.24 mM NADH, lactate dehydrogenase (10 U), 0.2 mM oxaloacetate, 0.5 mM oxamate, and 0.25 mM acetyl CoA. All reactions were initiated by the addition of oxaloacetate and performed in 1 mL quartz cuvettes. The final PC concentration ranged from 50 to 200 μg/mL (0.4–1.6 μM). All $k_{cat}$ values were calculated by dividing the experimentally determined $V_{max}$ values by the total protein concentration, defined as the subunit molar concentration (i.e., using a molecular weight of 126 kDa for RePC and 130 kDa for AnPC). Protein concentrations were determined spectrophotometrically by measuring the absorbance at 280 nm for wild-type and hybrid enzymes. The extinction coefficients for the fully reduced enzyme were calculated from the primary sequence.

**Generation of ΔBC and ΔBCΔBCCP RePC hybrid dimers.** To assess the activity of the ΔBC T882A/ΔBCΔBCCP RePC hybrid dimer, both ΔBC T882A and ΔBCΔBCCP RePC were diluted to 1 mg/mL in 20 mM Tris-HCl, pH 7.8; 200 mM NaCl and 6 mM β-mercaptoethanol. The two enzymes were mixed together at a 1:1 ratio. The ΔBC T882A, ΔBCΔBCCP, and ΔBC T882A/ΔBCΔBCCP RePC mixture, all at a final concentration of 1 mg/mL, were incubated for 2 h at room temperature before assessing their oxaloacetate decarboxylation activity, as described above, in both the presence and absence of 0.5 mM oxamate. All reactions were performed in 1 mL reaction volumes at 25 °C in 100 mM Tris-HCl, pH 7.8. The reaction was initiated by the addition of 150 μg of ΔBC T882A, ΔBCΔBCCP or the ΔBC T882A/ΔBCΔBCCP mixture.

To generate mixed combinations of the wild-type ΔBC RePC with the inactive CT dimer ΔBCΔBCCP T882A RePC, the procedure for remixing was as follows: ΔBC RePC and ΔBCΔBCCP T882A were diluted to 3 mg/mL in 100 mM Tris (pH 7.8) before being combined together at a given ratio to a final protein concentration of 1.0 mg/mL. The mixtures were incubated for 2 h at room temperature before assessing their oxaloacetate decarboxylation activity, as described above, in the presence of 0.5 mM oxamate. All reactions were performed in 1 mL reaction volumes at 25 °C in 100 mM Tris-HCl, pH 7.8. The reaction was initiated by the addition of 150 μg of ΔBC, ΔBCΔBCCP T882A or the ΔBC/ΔBCΔBCCP T882A mixture.

The relative recovered activity measured from the mixture comparing to RePC ΔBC dimer were plotted against the ratio (1:x) for ΔBC:ΔBCΔBCCP T882A. The equation describing the theoretical curve for a 100% *cis* translocation in the BCCP domain is shown in Eq. (1), while the equation describing 0% *cis* translocation in the BCCP domain is shown in Eq. (2). The derivation of these equations is provided in Supplementary Fig. 4. The experimental data were fit to an interpolation of Eqs. (1) and (2), as described in Eq. (3), where $n$ represents the fraction to which the data by a 100% *cis* translocation.

$$y = 1/(1 + x) \tag{1}$$

$$y = 1/(1 + x)^2 \tag{2}$$

$$y = n\left[\frac{1}{(1 + x)}\right] + (1 - n)\left[\frac{1}{(1 + x)^2}\right] \tag{3}$$

**Determination of $K_a$ for acetyl CoA and $K_i$ for L-aspartate**. The activity of pyruvate carboxylation was measured over a range of L-aspartate/acetyl CoA concentrations, in order to determine the $K_i$ for L-aspartate and the $K_a$ for acetyl CoA. The $K_a$ value for acetyl CoA was determined from a plot of the initial velocity (normalized for enzyme concentration) against acetyl CoA concentration and by fitting the data to Eq. (4) using least-squares nonlinear regression, where [AcoA] is the acetyl CoA concentration. All least-square fits were performed using Kaleida-Graph programs. $v_i/[E]_t$ is the initial velocity normalized for the enzyme concentration; $k_{cat}^0$ is the $k_{cat}$ value in the absence of acetyl CoA; $k_{cat}^{AcoA}$ is the $k_{cat}$ value at saturating concentrations of acetyl CoA; $K_a$ is the activation constant (apparent equilibrium dissociation constant) for acetyl CoA; and $h$ is the Hill coefficient with respect to acetyl CoA.

$$\frac{v_i}{[E]_t} = \left(k_{cat}^0 + k_{cat}^{AcoA} \times \left(\frac{[AcoA]}{K_a}\right)^h\right) \Big/ \left(1 + \left(\frac{[AcoA]}{K_a}\right)^h\right). \tag{4}$$

Similarly, the $K_i$ for L-aspartate was determined from a plot of the initial velocity (normalized for enzyme concentration) against L-aspartate concentration and fitting the data to Eq. (5) using least-squares nonlinear regression, where [Asp] is the L-aspartate concentration, $k_{cat}^0$ is the $k_{cat}$ value in the absence of L-aspartate; $k_{cat}^{asp}$ is the $k_{cat}$ value at saturating concentrations of L-aspartate; $K_i$ represents the inhibition constant for L-aspartate; and $h$ is the Hill coefficient with respect to L-aspartate.

$$\frac{v_i}{[E]_t} = \left(k_{cat}^0 + k_{cat}^{asp} \times \left(\frac{[Asp]}{K_i}\right)^h\right) \Big/ \left(1 + \left(\frac{[Asp]}{K_i}\right)^h\right) \tag{5}$$

**Data availability**. The data that support the findings of this study are available from the corresponding author upon reasonable request.

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

## Acknowledgements

This work was supported by the National Institutes of Health grant GM117540.

## Author contributions

Y.L. contributed most of the experiments, analyzed the results and wrote the paper. M.M.B. and K.S. contributed the experiments and analyzed the results for the ΔBC RePC dimers. M.St.M. contributed the idea for the project, and wrote the paper with Y.L.

## Additional information

**Competing interests:** The authors declare no competing interests.

