## [Peer Review File · Nature Communications]

PEER REVIEW FILE

Reviewers' Comments:

Reviewer #1 (Remarks to the Author):

Review of NCOMMS-18-02624-T

Title: Allosteric regulation alters carrier domain translocation in pyruvate carboxylase

Authors: Liu et al.

Summary

This is a revision of a manuscript in which the authors describe the mechanism by which acetyl-CoA allosterically activates the function of the assembly line enzyme, pyruvate carboxylase. The pyruvate carboxylase enzyme is a member of a large class of biotin-dependent carboxylases in which the three functional sites including the biotin carboxylase (BC), the biotin carboxyl carrier protein (BCCP) and transcarboxylase (TC) are encoded on different subunits (in prokaryotic enzymes). In the enzyme functional cycle the BCCP subunit is first carboxylated at the BC site and then translocates to the TC site for transfer of the carboxyl group to an acceptor. In textbook representations of this enzyme the linkage between the biotin moiety and BCCP is depicted as the "swinging arm" that enables transfer of the carboxylated biotin on BCCP from BC to TC. Structural and biochemical studies of these enzymes have revealed that the unstructured linker that joins the two BCCP domains, one of which participates in assembly and the second of which contains the lysine residue that is the target of biotinylation, is responsible for translocation of the carboxylated biotin from BC to TC. This manuscript defines the nature of intersubunit translocation in *R. etli* pyruvate carboxylase and how the allosteric activator influences translocation. Using reconstituted enzyme constructs in which the activities of specific subunits are either deleted or functionally compromised these authors elegantly demonstrate two novel features of the enzyme. First, in the absence of allosteric activator, acetylCoA, a BCCP subunit

has can access a number of TC sites. This result is surprising because recent structural studies suggested direct translocation to a single TC site. Second, in the presence of allosteric activator, acetylCoA, BCCP translocation is restricted to one path. The work is of general significance for understanding inter-domain movements in "assembly line" enzymes and their allosteric regulation. The authors have adequately addressed the reviewers' concerns and the manuscript is acceptable for publication.

I have a few suggestions for minor changes:

1. Figure 3, legend, line 2: " $\Delta ABC\Delta BCCCP$ " should be " $\Delta ABC\Delta BCCP$ "
2. Discussion, page 19, lines 6 and 7: Consequently is used on consecutive sentences, which is repetitious.
3. Experimental procedure, page 24, Do the authors have a reference for how protein extinction coefficients were calculated?
4. Page 27, Should hill coefficient be Hill coefficient?
5. Page 27, Why is K_a for acetyl CoA referred to as activation constant? Isn't it an apparent equilibrium association constant?
6. Page 27, Equations (4) and (5): I think that some subscripts are missing.

Reviewer #2 (Remarks to the Author):

The authors have clarified the points raised earlier, and the manuscript is appropriate for Nature Communications.

A recent structure of *L. lactis* PC suggests a different BCCP translocation pathway, which appears to be equivalent to pathway c in this manuscript. It would be helpful if this is described in the paper.

Responses to Reviewers' Comments:

We thank the reviewers for their evaluation of our revised manuscript. They have provided a few suggestions for minor revisions. Our responses to these suggestions are addressed below. The related editorial changes to the manuscript are highlighted in red.

Reviewer #1 (Remarks to the Author):

Summary

This is a revision of a manuscript in which the authors describe the mechanism by which acetyl-CoA allosterically activates the function of the assembly line enzyme, pyruvate carboxylase. The pyruvate carboxylase enzyme is a member of a large class of biotin-dependent carboxylases in which the three functional sites including the biotin carboxylase (BC), the biotin carboxyl carrier protein (BCCP) and transcarboxylase (TC) are encoded on different subunits (in prokaryotic enzymes). In the enzyme functional cycle the BCCP subunit is first carboxylated at the BC site and then translocates to the TC site for transfer of the carboxyl group to an acceptor. In textbook representations of this enzyme the linkage between the biotin moiety and BCCP is depicted as the "swinging arm" that enables transfer of the carboxylated biotin on BCCP from BC to TC. Structural and biochemical studies of these enzymes have revealed that the unstructured linker that joins the two BCCP domains, one of which participates in assembly and the second of which contains the lysine residue that is the target of biotinylation, is responsible for translocation of the carboxylated biotin from BC to TC. This manuscript defines the nature of intersubunit translocation in *R. etli* pyruvate carboxylase and how the allosteric activator influences translocation. Using reconstituted enzyme constructs in which the activities of specific subunits are either deleted or functionally compromised these authors elegantly demonstrate two novel features of the enzyme. First, in the absence of allosteric activator, acetylCoA, a BCCP subunit has can access a number of TC sites. This result is surprising because recent structural studies suggested direct translocation to a single TC site. Second, in the presence of allosteric activator, acetylCoA, BCCP translocation is restricted to one path. The work is of general significance for understanding inter-domain movements in "assembly line" enzymes and their allosteric regulation. The authors have adequately addressed the reviewers' concerns and the manuscript is acceptable for publication.

I have a few suggestions for minor changes:

1. Figure 3, legend, line 2: " $\Delta BC\Delta BCCCP$ " should be " $\Delta BC\Delta BCCP$ "

We have made this correction in the figure legend for Figure 3.

2. Discussion, page 19, lines 6 and 7: Consequently is used on consecutive sentences, which is repetitious.

We agree that this repetition is awkward. We have changed "Consequently" to "Thus" in the second sentence.

3. Experimental procedure, page 24, Do the authors have a reference for how protein extinction coefficients were calculated?

A reference (reference #33) has been added that describes the ExPASy ProtParam software that was used to calculate the extinction coefficients.

4. Page 27, Should hill coefficient be Hill coefficient?

We agree. This should be “Hill”, and not “hill”. We have made this change throughout.

5. Page 27, Why is K_a for acetyl CoA referred to as activation constant? Isn't it an apparent equilibrium association constant?

Here we disagree. The K_a value *is* best referred to as an activation constant, which is defined as the equilibrium *dissociation constant* (note that the units are expressed in concentration) for the activator. To reduce confusion, we refer to this as the activation constant and, in brackets, describe this as an “apparent equilibrium dissociation constant”.

6. Page 27, Equations (4) and (5): I think that some subscripts are missing.

The reviewer doesn't specify which subscripts are missing, so we presume that the reviewer is suggesting the use of v_i (instead of v) and $[E]_t$ (instead of $[E]$). We have added these subscripts to the equations and have also added them these variables in the text.

Reviewer #2 (Remarks to the Author):

The authors have clarified the points raised earlier, and the manuscript is appropriate for Nature Communications.

A recent structure of L. lactis PC suggests a different BCCP translocation pathway, which appears to be equivalent to pathway c in this manuscript. It would be helpful if this is described in the paper.

We thank the author for drawing this most recent structure and paper to our attention. This article was published after our initial submission and we initially missed incorporating its highly complementary findings into our revised manuscript. A discussion of this paper and structure has been added to the opening paragraph of our discussion (page 11)